# Long-Term Follow-Up of Intensive Integrative Treatment including Motion Style Acupuncture Treatment (MSAT) in Hospitalized Patients with Lumbar Disc Herniation: An Observational Study

**DOI:** 10.3390/healthcare10122462

**Published:** 2022-12-06

**Authors:** Mu-Jin Park, So-Ri Jin, Eun-Song Kim, Hyun-Seok Lee, Kyu-Hyun Hwang, Seung-Ju Oh, Jee Young Lee, Doori Kim, Yoon Jae Lee, In-Hyuk Ha

**Affiliations:** 1Department of Korean Internal Medicine, Bundang Jaseng Hospital of Korean Medicine, Sungnam-si 13590, Republic of Korea; 2Departments of Acupuncture and Moxibustion, Bundang Jaseng Hospital of Korean Medicine, Sungnam-si 13590, Republic of Korea; 3Department of Korean Rehabilitation Medicine, Bundang Jaseng Hospital of Korean Medicine, Sungnam-si 13590, Republic of Korea; 4Department of Korean Internal Medicine, Integrative Cancer Center, Cha Ilsan Medical Center, 1205, Jungang-ro, Ilsandong-gu, Goyang-si 10444, Republic of Korea; 5Jaseng Spine and Joint Research Institute, Jaseng Medical Foundation, 2F, 540 Gangnam-daero, Gangnam-gu, Seoul 06110, Republic of Korea

**Keywords:** motion style acupuncture treatment, integrative treatment, lumbar disc herniation, surveys and questionnaires

## Abstract

This study aimed to investigate the long-term effects of and satisfaction with integrative Korean medicine treatment and motion style acupuncture treatment (MSAT) in patients with lumbar disc herniation (LDH). We retrospectively analyzed medical charts and prospectively surveyed adult patients aged between 19 and 64 years treated for lumbar disc herniation for at least 6 days at three Korean hospitals from 1 January 2015 to 31 December 2020. The primary outcome was the Numeric Rating Scale (NRS) for back pain. Secondary outcome measures included the NRS for radiating leg pain, the Oswestry Disability Index (ODI), and the European Quality of Life-5 Dimension-5 Level (EQ-5D-5L) questionnaire. The NRS scores for low back pain decreased from 5.40 ± 1.58 to 2.92 ± 2.09, NRS for radiating leg pain from 5.57 ± 1.56 to 1.78 ± 2.36, and ODI from 46.39 ± 16.72 to 16.47 ± 15.61 at baseline and survey, respectively. The EQ-5D-5L increased from 0.57 ± 0.19 to 0.82 ± 0.14. In conclusion, Korean medicine and MSAT could be effective treatment methods for patients with LDH. The results of this study can be used as helpful information for clinicians who treat patients with LDH in real clinical settings.

## 1. Introduction

Low back pain (LBP) is a common condition that at least 70% of the population experience at least once during their lifetime [1]. There are various causes of LBP; however, the most common is lumbar disc herniation (LDH) [2]. LDH is characterized by herniation of a ruptured fibrosus annulus of the disc that compresses the spinal cord or nerve roots, causing ischemia and inflammation [3]. Along the compressed nerves, radiating pain, paresthesia, and muscle weakness may occur in the lumbar spine and lower limb regions [4].

As LDH becomes more severe, symptoms and dysfunction also tend to worsen, but they are not directly proportional [5]. The clinical symptoms should also be considered. In general, the incidence of LDH is approximately 20 cases per 1000 adults annually and it is most common in people in their third to fifth decades of life [6]. More men are diagnosed with LDH than women, with a male-to-female ratio of 3:2 [7]. The prevalence of LDH is approximately 1–3% in Italy and Finland and 1–2% in the USA, with an overall prevalence of >1% [8]. Regarding the incidence of LDH by spinal level, the most common levels are L4/5 (approximately 40–45%), followed by L5/S1 (approximately 30–35%) and the remaining levels (around 10% each). Multilevel LDHs account for ≥10% of cases [9,10], and LDHs at levels higher than L4 are more common in people aged ≥55 years [11].

Most patients with symptoms of acute LDH show favorable results with improvement over 2–3 months, whereas some require additional tests and treatments [12]. The American College of Physicians (ACP) recommends the use of appropriate rest and conservative therapy to treat LBP and LBP with radiating leg pain. In general, non-pharmacological therapies recommended by ACP guidelines consist of superficial heat therapy, massage along with acupuncture, and spinal manual therapy. Additionally, guidelines recommend that clinicians concurrently use exercise, rehabilitation treatment, and stress reduction therapy [13]. Moreover, acupuncture is commonly used to treat LDH in many countries, including Korea and China, and relevant clinical trials have been conducted [14].

Motion style acupuncture treatment (MSAT) is an acupuncture treatment combined with Daoyin exercise therapy, resulting in complex effects by exposing patients to passive and active movements with the needle inserted [15]. MSAT is classified as T-MSAT [16] when using a traction device and H-MSAT [17] when using a number of medical staff. MSAT is mainly performed for immobile patients due to acute LDH or sprain and has significant effects on recovery and pain reduction [17,18]. A previous multicenter randomized controlled trial (RCT) [17] that evenly allocated 58 patients to an MSAT group or diclofenac sodium injection treatment group reported that the MSAT group showed better pain reduction and recovery of abilities in the first month.

Although the aforementioned treatments do not include pharmacological interventions, and side effects from interactions between treatments are less concerning, previous studies have been conducted to demonstrate the efficacy of each treatment method rather than the efficacy of integrative treatment [19,20,21,22]. However, multiple complex treatments are currently used in clinical settings, and few studies have reported the effectiveness and safety of long-term use of high-intensity complex treatments.

The rehabilitation and early mobility of patients with LDH has been reported; however, further studies assessing the efficacy of MSAT in terms of helping mobility with assistance combined with acupuncture in real-time are required. This study aimed to investigate the efficacy of high-intensity Korean medicine integrative treatment in patients with LDH in improving lumbar spine function and to report the degree and necessity of non-pharmacological integrative Korean medicine treatment by analyzing subgroups who additionally received MSAT.

Therefore, this study aimed to share the clinical results of diverse and aggressive integrative Korean medicine treatments, including MSAT, in patients with LDH who visited and were treated at Jaseng Hospital of Korean Medicine.

## 2. Materials and Methods

### 2.1. Study Design

This observational study was based on a retrospective chart review and long-term follow-up. This study was conducted in accordance with the Strengthening the Reporting of Observational Studies in Epidemiology (STROBE) guidelines.

From 1 January 2015 to 31 December 2020, patients who were diagnosed with LDH and hospitalized in three Jaseng Hospitals of Korean medicine (in Gangnam, Bundang, and Ulsan in Korea), spine, and joint hospitals were included. Among the participants who met the inclusion/exclusion criteria, those who agreed that their personal information could be used for study purposes at the time of admission were included in the study.

Demographic characteristics and relevant medical records of participants who met the inclusion and exclusion criteria were collected from the electronic medical records (EMR). Subsequently, a follow-up questionnaire survey was conducted from February 2022 to April 2022. The survey was conducted in two ways: an online survey using Google questionnaire and a phone survey. First, an online questionnaire was administered. If the participants did not respond, a telephone survey was conducted. If participants responded to the online survey, a telephone survey was not conducted. For the telephone survey, a phone call was made on three different dates until the participants answered. If the participants did not answer the telephone survey three times or clearly expressed the intention to refuse the survey via phone call, they were classified as non-respondents. This study was reviewed and approved by the Institutional Review Board (IRB) of the Jaseng Hospital of Korean Medicine (File No.: JASENG 2021-12-010; approval date: 20 December 2021), and all investigators complied with the Declaration of Helsinki.

### 2.2. Participants

#### 2.2.1. Inclusion Criteria

Patients who met all of the following criteria were eligible for participation: (1) male and female adults aged between 19 and 64 years of age; (2) received a diagnosis of LDH with clinical or radiological documentation, and (3) at admission, expressed chief complaint of LBP or radiating leg pain with a numeric rating scale (NRS) score of 4 or higher.

#### 2.2.2. Exclusion Criteria

The exclusion criteria of this study were as follows: (1) history of lumbar spine diseases including stenosis, malignant tumors, and soft tissue diseases that are known to markedly affect LBP and radiating leg pain; (2) history of lumbar spine region surgery or trauma that notably affects LBP and radiating leg pain within 3 months before the creation of the medical records; (3) hospitalized for not more than 6 days; (4) had underlying disease that may affect orientation to time, place, and person, or not able to comply with the questionnaire survey for any other reasons; (5) had no relevant assessment records because the participant refused lumbar spine region assessment during the hospital stay; (6) if the investigator considered the participant to be unsuitable for any reason to participate in the study.

### 2.3. Data Collection

Study data were analyzed mainly for length of hospital stay, time of follow-up, and especially for the time of admission, discharge, and follow-up. Based on medical records during the hospital stay, demographic characteristics (sex, age, and occupation), medical history, social history, and underlying diseases (hypertension, diabetes, depression, heart disease, respiratory disease, and gastrointestinal disease) were investigated. The medical records of the patients with LBP were collected. Chief complaint; length of hospital stay; onset date; origin of disease; current medical history; physical, hematological, and radiological diagnosis including magnetic resonance imaging (MRI) findings; pain severity and function tests performed during hospital stay; and evaluation information, such as health-related quality of life, if there was one related to disease, were collected.

History of prescriptions was analyzed to investigate treatment items that the participants received during the hospital stay and to calculate the mean number of treatments by treatment item. In the case of herbal medicines, the number of doses was investigated, and the number of treatments was investigated for other treatment items. Treatment methods included MSAT, acupuncture, electroacupuncture, pharmacopuncture, cupping therapy, herbal medicine, Chuna manual therapy (CMT; the manipulative therapy of Korean Medicine), and other physical therapies in Korean Medicine. Adverse reactions that occurred during inpatient treatment were documented.

### 2.4. Long-Term Follow Up Survey

The questionnaire survey consisted of information about the status of treatment after discharge and type of treatment; diagnosis of other lumbar diseases and history of surgery after discharge; NRS for measuring lumbar pain and radiating leg pain; Oswestry Disability Index (ODI) for measuring disability of the lumbar region; European Quality of Life-5 Dimension-5 Level (EQ-5D-5L) for measuring health-related quality of life, and the Patients’ Global Impression of Change (PGIC) for measuring satisfaction with and improvement associated with Korean medicine treatment. The questionnaire was developed through agreement between the investigators, and information about the history of treatment and surgery from the end of the treatment to the time of completing the questionnaire was identified to determine the basic condition. Moreover, the same pain, disability, and quality of life indices were used before and after treatment for consistency in analyzing the treatment effects. The treatment types used in the questionnaire were largely divided into Korean and Western medicine treatments. Korean medicine treatments included herbal medicine, acupuncture, cupping therapy, CMT, pharmacopuncture, and moxibustion. Western medicine treatment included analgesics, physical therapy, rehabilitation therapy, and injection treatment. The number of treatments was then determined. In addition, the type of treatment(s) that the patients were satisfied with and the reason(s) for satisfaction or dissatisfaction with the treatment(s) during the hospital stay were investigated for development and improvement through feedback, and the PGIC scores were collected and evaluated. The PGIC [23] is a 7-point scale depicting a patient’s participative rating of overall improvement after treatment (1, completely improved; 2, much improved; 3, minimally improved; 4, no change; 5, minimally worse; 6, much worse; and 7, very much worse). To evaluate satisfaction with integrative Korean medicine treatment, the PGIC was administered via survey, and improvement in pain and discomfort after discharge was evaluated.

### 2.5. Intervention

Interventions used for relieving pain and recovering the ability of patients during their hospital stay included the treatments discussed below.

#### 2.5.1. MSAT

##### H-MSAT

For the H-MSAT, two physician assistants stood on both sides of the patient with their arms around the patient’s waist while gently holding one of the patient’s hands. In this position, the practitioner inserted stainless steel disposable needles (0.25 × 30 mm; Dong-bang Medical, Seongnam, Korea) to a depth of 10–15 mm at the participant’s Pungbu (Governor Vessel Meridian 16; GV16) and on both sides of Haenggan (Liver Meridian 2; LR2) and Gokji (Large Intestine Meridian 11; LI11). These acupuncture points were selected according to the traditional Korean medicine theory (qi circulation) and previous clinical experience. The location of each acupuncture point was determined using the guidelines published by the World Health Organization (WHO) Standard Acupuncture Point Locations in the Western Pacific Region. The patient was asked to walk with the assistance of the medical staff with the needles retained at the acupoints. As the patient’s walking ability improved and pain was relieved, the patient was asked to walk by themselves, and medical staff assisted on both sides gradually stopped supporting the participant. When the patient could walk without any support, all the needles were removed, and the patient was asked to continue walking for another 1–2 min. The medical staff encouraged the patient by numbering the patient’s steps to distract the patient’s attention, move seamlessly, and support the patient to reduce the patient’s apprehension of movement. This procedure took approximately 20 min to complete.

The H-MSAT technique was considered in patients with pain severe enough to cause dysfunction in terms of mobility and ambulation. However, the use of H-MSAT was still at the physician’s discretion.

##### T-MSAT

Unlike the H-MSAT, which requires two physician assistants, the T-MSAT uses traction to help patients walk. For retraction, de-weight balance retraction (GEM-TECH, Siheung, Korea) of an electric orthopedic exercise device (GEM-TECH, Siheung, Korea) was used. The acupoints and needling methods used were the same as those of the H-MSAT; with the help of traction with a weight of 50% of the patient’s weight, the patient was asked to walk and was supervised. The patient was asked to walk 10 times for 10 to 15 min in a space of 10 m by making round trips. If the patient was unable to walk well, depending on the patient’s condition, the walking pace was reduced appropriately, and the procedure was performed for approximately 10 to 15 min. The medical staff encouraged the patient by numbering the patient’s steps to distract the patient’s attention, move seamlessly, and support the patient to reduce the patient’s apprehension of movement.

Similar to H-MSAT, T-MSAT was performed in patients with severe pain that limited movement. However, it was in place of H-MSAT when assistant doctors were unavailable during therapy.

#### 2.5.2. Acupuncture and Electroacupuncture

The practitioner used stainless steel disposable needles (0.25 × 30 mm, Dongbang Medical, Seongnam, Korea) for acupuncture. Based on the judgment of the clinician performing acupuncture, the following acupoints were selected around the lumbar and lower limb regions: Sinus, Bladder Meridian 23 (BL23), Zhilbian, Bladder Meridian 54 (BL54), Sameumgyo, Spleen Meridian 6 (SP6), Hyeonjong, Gall Bladder Meridian 39 (GB39), and Myungmoon, Governor Vessel Meridian 4 (GV4). Needles were inserted to a depth of 0.5 to 1.0 cm for 15 min twice a day.

StraTek STN-111 (StraTek, Anyang, Korea) was used as the electrical stimulator. Electroacupuncture therapy was performed twice daily, similar to acupuncture therapy.

#### 2.5.3. Pharmacopuncture

For pharmacopuncture therapy, a disposable insulin syringe (29 G × 13 mm, 1 mL, Sungshim Medical, Bucheon, Korea) was used to inject 1–2 mL of herbal medicine to the following acupoints: Sinus, Bladder Meridian 23 (BL23), Zhilbian, Bladder Meridian 54 (BL54), Myungmoon, Governor Vessel Meridian 4 (GV4), Jisil, Bladder Meridian 52 (BL52), and Yoyanggwan, Governor Vessel Meridian 3 (GV3). The pharmacopuncture solutions included ShinBaro pharmacopuncture (Jaseng Spine and Joint Research Institute, Namyangju, Korea) and bee-venom pharmacopuncture at 5% and 10% (Jaseng Spine and Joint Research Institute, Namyangju, Korea). Pharmacopuncture was administered twice daily.

#### 2.5.4. Chuna Manual Therapy (CMT)

The manipulative therapy of Korean Medicine, CMT was performed with the Korean medicine doctor’s determination for approximately 10 to 15 min once daily, depending on each patient’s symptoms and progression. CMT is a manipulative treatment method used to recover all structural abnormalities of the spine and joint system to normal function and structure; it is administered by a Korean medicine doctor using the hands, body parts, or other assistive devices such as tables and stimulation through passive exercise or correcting displacement. Currently, CMT is a treatment method that combines excellent manipulative therapy from China, Japan, India, and the USA based on the traditional Korean medicine technique according to the Korean medicine theory. CMT is used to treat patients with musculoskeletal and neuromuscular diseases. In Korea, CMT is combined with supportive therapy for lumbar pain, and among the existing studies on lumbar pain, most have been conducted on disc herniation [24].

#### 2.5.5. Herbal Medicine

The patients were instructed to take 3 packs/2 potions of herbal medicine three times daily, 30 min after meals. Herbal medicine was prescribed according to each patient’s characteristics, symptoms, and progress. The herbal medicine containing GCSB-5 as its main ingredient is known to be effective, especially in relieving nerve stimulation due to LDH with anti-inflammatory, neuroregenerative, and analgesic effects, as demonstrated by an experimental study [25] and clinical trial [26]. Depending on the patient’s condition, drugs that improved blood circulation (Hwalhyul) and reduced pain (Jitong) were concomitantly administered.

#### 2.5.6. Other Treatments

Herbal hot pack therapy has been used in Korean medicine for physiotherapy. A heating pad steamed with any of the following 23 herbal drugs with anti-inflammatory and analgesic effects was used: Achyranthes bidentata Blume, Saposhnikoviae Radix, and Angelicae Dahuricae Radix. The heating pad was placed on the lumbar region once daily for 15–20 min. Cupping therapy was performed at two painful sites in the lumbar spine region, avoiding the needle point. Disposable sterile cupping cups (size 3; Dongbang Medical, Seongnam, Korea) were used. Cupping therapy was performed for 15 min concurrently with acupuncture therapy. The cupping therapy was administered twice daily.

If necessary, Western medicine treatments, such as manipulative therapy and physical therapy, were performed in cooperation with a Western medicine doctor.

### 2.6. Outcome Measures

#### 2.6.1. Primary Outcome

##### NRS for LBP 

The NRS is a numeric pain screening tool that objectively converts the participative intensity of pain. The scale ranges from 0 (no pain) to 10 (worst pain imaginable). The patients were asked to select one number that fit best their pain severity. LBP severity was also measured. Data on NRS at admission and discharge were collected through EMR, and the patient was asked to complete NRS after discharge to measure the severity of their pain at that moment in time [27].

#### 2.6.2. Secondary Outcomes

##### NRS for Radiating Leg Pain 

The severity of radiating leg pain was also measured using the NRS. Data on NRS at admission and discharge were collected through EMR, and the patient was asked to complete NRS after discharge [27].

##### Lumbar Region Disability Index (ODI)

The ODI [28] is a questionnaire developed by Fairbank that has been designed to evaluate functional disability in daily living in patients with LBP. In 2005, The Korean version of the ODI was validated [29]. It is a self-reported questionnaire that consists of the following 10 items that mark the level of discomfort or dysfunction perceived by the patient: pain intensity, personal care, lifting, walking, sitting, standing, sleeping, sex life, social life, and travelling. Each item is scored on a 0–5 rating scale and calculated in terms of 100 percentiles based on the responses provided. In this study, the scores evaluated on the day of admission and discharge were collected, and further scores, which were evaluated at 15 days of hospital stay, were collected from participants who had a hospital stay of ≥15 days. Like NRS, participants were requested to get ODI done after discharge by survey.

##### EQ-5D-5L Score

The EQ-5D-5L is designed to measure health-related quality of life and has been widely used in the healthcare field. The EQ-5D-5L consists of five dimensions: mobility, self-care, usual activities, pain/discomfort, and anxiety/depression. Each dimension assesses ability on five levels (level 1, no problems; level 2, mild problems; level 3, moderate problems; level 4, severe problems, and level 5, extreme problems). In this study, EQ-5D-5L was calculated by applying the weighted model estimated for the Korean population. Data on EQ-5D-5L evaluated on the days of admission and discharge were collected through the EMR, and a questionnaire survey was conducted to collect current data on EQ-5D-5L [30].

### 2.7. Statistical Analysis

The analysis set included data from retrospective medical records and a prospective questionnaire survey. A two-tailed test was used to perform the statistical analysis, and the significance level was set at 5%. For the basic characteristics of the analysis set, continuous variables are presented as mean ± standard deviation (SD). Categorical variables are presented as descriptive analyses in terms of frequency and percentage. With respect to the *p*-value, a chi-squared test or Fisher’s exact test was performed for continuous variables, while an independent *t*-test was used to analyze continuous variables. For the history of treatments, the status of treatment is represented as the number of participants and percentage, and differences between the two groups were analyzed using the chi-squared test or Fisher’s exact test. The number of treatments is expressed as the mean ± SD, and the difference between the two groups was analyzed using an independent *t*-test.

The values of outcomes (NRS for LBP, NRS for radiating leg pain, ODI, and EQ-5D-5L) measured at admission, discharge, and follow-up are expressed as the mean ± SD. The change from baseline is presented as a 95% confidence interval (CI) using a linear mixed model. In the linear mixed model, participants were included as random effects, and time was included as a fixed effect. Time was calculated using categorical variables. The baseline outcomes were also corrected. In addition, to calculate the proportion of change at each time point, a linear mixed model including the cross terms of time and MSAT status was utilized. The changes in outcome values from baseline to discharge and follow-up of each group are expressed as the least square means and 95% CI, which were estimated using a linear model. The baseline outcomes of the two groups are expressed as the mean ± SD.

Survival analysis was conducted for the NRS score for LBP, which was the primary outcome. Recovery from LBP was defined as a 2.5-point reduction in substantial clinical benefit (SCB) on the NRS. The Kaplan–Meier survival curve was used for survival analysis, and the difference between the MSAT and non-MSAT groups was compared using the log-rank test. In addition, a Cox proportional hazards model was used to calculate the hazard ratio for the recovery of the two groups.

Factors affecting recovery from LBP were analyzed using a logistic regression model. Univariate and multivariate logistic models were used to evaluate the influencing factors, including baseline outcomes, MSAT status, sex, age, body mass index (BMI), alcohol consumption, smoking, occupation, length of hospital stay, and MRI findings. Odds ratios (OR) and 95% CIs are presented for each variable, and the area under the curve (AUC) value was presented for the multivariate regression model.

## 3. Results

### 3.1. Study Flow

This study involved the administration of a questionnaire survey to 435 patients who met the eligibility criteria among those diagnosed with LDH and were hospitalized in three integrative hospitals of Western and Korean Medicine located in Korea from 1 January 2015 to 31 December 2020. To conduct follow-up surveys, patients who refused to receive text message were excluded right at the beginning of the study. The participants were allocated to either the MSAT group (who had MSAT) or the n-MSAT group (who did not have MSAT) for analysis. There were 80 participants in the MSAT group and 355 participants in the n-MSAT group. A total of 152 participants completed the questionnaire (28 in the MSAT group and 124 in the n-MSAT group; Figure 1).

### 3.2. Baseline Characteristics

The mean age of the participants was 45.22 ± 12.67 years, and 78 participants (51.32%) were men. The overall mean BMI was 24.78 ± 3.98, while the mean BMI in the MSAT group and n-MSAT group was 23.26 ± 3.10 and 25.12 ± 4.09, respectively (*p* = 0.025). The mean length of hospital stay was 26.09 ± 17.56, 34.93 ± 21.82, and 24.10 ± 15.89 days in both groups, the MSAT group, and the n-MSAT group, respectively, which shows that the participants in the MSAT group had longer treatment duration (*p* = 0.018). At admission, the NRS scores for radiating leg pain in the MSAT and n-MSAT groups were 6.14 ± 1.35 and 5.44 ± 1.58 in the MSAT group and n-MSAT group, respectively, indicating that the MSAT group had more severe pain (*p* = 0.032). For LBP, NRS, ODI, and EQ-5D-5L scores tended to be similar. Protrusion was observed in all participants, while extrusion and sequestration were observed in 67 (44.08%) and 9 participants (5.92%), respectively, based on MRI with radiologist interpretation (Table 1).

### 3.3. Treatments

Korean medicine treatment performed during the hospital stay included acupuncture and electroacupuncture, pharmacopuncture, bee venom acupuncture, herbal medicine (general herbal medicine and GCSB-5-based herbal medicine), and Chuna manual therapy. Western medicine treatment included manipulative therapy and physical therapy (including traction and extracorporeal shock wave therapy). All patients underwent acupuncture and electroacupuncture treatments, and the mean number of treatments performed was 87.51 ± 60.91. Pharmacopuncture and bee venom acupuncture were performed for all participants in the MSAT group and 115 (92.74%) participants in the n-MSAT group. In the MSAT group, 85.71% of participants received herbal medicine containing GCSB-5, with a mean number of treatments performed of 27.18 ± 20.61. In the n-MSAT group, 87.10% of participants received GCSB-5 based herbal medicine, with a mean number of treatments performed of 19.18 ± 16.63. In both groups, 135 participants (88.82%) underwent Chuna manual therapy, with a mean number of treatments performed of 15.84 ± 11.70. More patients in the MSAT group underwent each treatment. The proportion of patients receiving acupuncture and electroacupuncture, pharmacopuncture and bee venom acupuncture, herbal medicine containing GCSB-5, Chuna manual therapy, and manipulative therapy was significantly higher in the MSAT group (Table 2).

### 3.4. Outcome Changes

NRS for LBP, NRS for radiating leg pain, ODI, and EQ-5D-5L scores for LBP and radiating leg pain measured at admission, discharge, and with the surveys were analyzed and showed that both the MSAT and n-MSAT groups demonstrated statistically significant improvements in all post-treatment outcomes (*p* < 0.001). The NRS score for LBP decreased by 2.72 (95% CI 2.45 to 2.98; *p* < 0.001) points, from 5.40 ± 1.58 points (at admission) to 2.68 ± 1.12 (at discharge after treatment). At follow-up, the score was 2.92 ± 2.09, showing a 2.48-point (95% CI 2.22 to 2.74; *p* < 0.001) reduction compared to the time of admission. The NRS score for radiating leg pain decreased by 2.74 (95% CI 2.44 to 3.04; *p* < 0.001), from 5.57 ± 1.56 points (at admission) to 2.83 ± 1.30 points (at discharge). The score was 1.78 ± 2.36 at follow-up, showing a 3.79-point (95% CI 3.49 to 4.09; *p* < 0.001) reduction compared to the time of admission (Table 3).

The NRS scores for LBP in the MSAT and n-MSAT groups were compared and analyzed. The change in the NRS scores in the MSAT group from baseline to discharge and follow-up was 0.14 points (95% CI −0.54 to 0.82; *p* = 0.693) and 0.03 points (95% CI −0.65 to 0.71; *p* = 0.924) higher, respectively. The change in the ODI score at discharge was 1.89 (95% CI −3.90 to 7.67; *p* = 0.522) points higher in the MSAT group, whereas the change in the ODI score at follow-up was 1.11 (95% CI −4.66 to 6.87; *p* = 0.706) points higher in the n-MSAT group. For all indicators, the differences in the changes between the groups were not significant (Table 4, Figure 2).

### 3.5. Survival Analysis

This study used SCB to determine improvement in pain, and the SCB for the NRS score for LBP was set to 2.5 points. The score that reached this number was considered significantly improved. Survival analysis was performed for achievement of SCB for NRS score for LBP, and the median survival times in the MSAT and n-MSAT groups were 56 (95% CI 31 to NA) days and 529 (95% CI 43 to 1205) days, respectively. The log-rank test result was 0.74 with a hazard ratio of 1.09 (95% CI 0.70 to 1.80; *p* = 0.74), indicating no significant difference between the two groups (Figure 3).

### 3.6. Follow-Up Survey

The mean duration from the time of patient admission to follow-up was 1141.50 (95% CI 844.50 to 1396.50) days, and there was no significant difference between the two groups. After discharge, one (3.57%) participant in the MSAT group and nine (7.26%) participants in the n-MSAT group underwent surgery for LDH, whereas 20 (71.43%) in the MSAT group and 65 (52.42%) in the n-MSAT group underwent non-surgical treatment.

During the hospital stay, most participants were satisfied with acupuncture (67 participants; 44.08%), acupuncture and electroacupuncture (67 participants; 44.08%), pharmacopuncture and bee venom acupuncture (67 participants; 44.08%), and Chuna manual therapy (61 participants; 40.13%). In the MSAT group, 15 participants (53.57%) were satisfied with the Chuna manual therapy. In the n-MSAT group, most participants were satisfied with acupuncture, acupuncture, electroacupuncture, pharmacopuncture, and bee venom acupuncture. In the MSAT group, 17.86% of the participants were satisfied with electroacupuncture, while in the n-MSAT group, only 5.65% of the participants were satisfied (*p* = 0.046).

The reasons for satisfaction and dissatisfaction with treatment during the hospital stay were analyzed. Most participants (85 participants; 55.92%) were satisfied with the treatment because they experienced a “significant reduction of pain.” In the MSAT group, most participants (five participants; 17.86%) were dissatisfied with the treatment because the “treatment duration was much longer than they expected.” In the n-MSAT group, most participants (24 participants; 19.35%) were dissatisfied with the treatment because the “treatment was not covered by medical insurance and costs were a burden”.

The PGIC measurements showed that symptoms had improved in 92.76% of respondents, with 45 (29.61%), 69 (45.39%), and 27 (17.76%) participants responding that their symptoms were “completely improved,” “much improved,” and “minimally improved,” respectively.

### 3.7. Factors Associated with Improvement

Factors affecting recovery from NRS of LBP at long-term follow-up were analyzed using logistic regression. The factors were evaluated using univariate and multivariate logistic regression models. In the univariate model, the odds for recovery were high when the baseline outcome was high, the participant underwent MSAT, was aged ≥ 50 years, drank alcohol, and experienced longer length of stay. A significantly high OR was observed in the baseline outcome (OR 3.01, 95% CI 2.07 to 4.40). In the multivariate model, the odds of recovery were high when the baseline outcome was high, the participant was at least 50 years old, BMI exceeded 25, the participant stayed in the hospital for at least 15 days, and had extrusion or sequestration on MRI. The OR was low when the participant was a woman. In the multivariate model, baseline outcomes (3.42, 95% CI 2.24 to 5.21) and female sex (0.27, 95% CI 0.08 to 0.91) were also significant. The AUC for the multivariate model was 0.883 (Table 5).

### 3.8. Safety

There were no adverse events (AEs) identified for MSAT and integrative Korean medicine treatments when reviewing the medical records.

## 4. Discussion

This study investigated the treatment effects and satisfaction through a prospective observational study in patients who were diagnosed with LDH and were hospitalized to receive integrative Korean medicine treatment in three integrative hospitals of Western and Korean Medicine located in Korea from 1 January 2015 to 31 December 2020. The analysis showed that among 152 respondents, 78 (51.32%) were men, slightly more than women. Most participants (89 participants; 58.55%) were aged ≤50 years. The mean BMI was 24.78 ± 3.98, which was between the normal weight range (18.5–24.9 kg/m^2^) and overweight range (25–29.9 kg/m^2^) of the BMI classification presented by WHO [31]. This result supports a previous study reporting that being overweight is a risk factor for LDH [32]. Weight (*p* = 0.034) and BMI (*p* = 0.025) showed a statistically significant difference between the two groups. This indicates that being overweight increases the risk of developing LDH and affects its severity.

The length of hospital stay was approximately 1.4-fold longer in the MSAT group than in the n-MSAT group (*p* = 0.018). All outcomes in the MSAT group at admission were worse than those in the non-MSAT group. In particular, the radiating pain showed a statistically significant difference (*p* = 0.032) between the two groups. This indicates that the severity of the condition was higher in the MSAT group than in the n-MSAT group. In a previous study [17], MSAT was performed in patients with severe disease. This study showed that MSAT can also be performed in patients with serious symptoms. The NRS scores for LBP and radiating leg pain in the participants significantly decreased at discharge. In particular, the score for radiating leg pain at follow-up was at least 1 point lower than that at discharge, showing that treatment effects lasted for a long period. In addition, the difference in the change in outcomes after treatment was greater in the MSAT group. The NRS for LBP, NRS for radiating leg pain, and ODI scores were high at admission in the MSAT group, but low at discharge and follow-up. However, no statistically significant differences were observed between the two groups.

The frequency of treatment performed during the hospital stay was analyzed, and it was found that all patients were treated with acupuncture and electroacupuncture. Pharmacopuncture, bee venom acupuncture, Chuna manual therapy, and herbal medicine were also frequently administered. As the MSAT group had a prolonged hospital stay, the number of sessions for acupuncture and electroacupuncture (*p* = 0.006), pharmacopuncture and bee venom acupuncture (*p* = 0.002), herbal medicine containing GCSB-5 (*p* = 0.03), Chuna manual therapy (*p* = 0.039), and manipulative therapy (*p* = 0.037) also showed a statistically significant difference between the MSAT and n-MSAT groups.

Many studies have investigated the effects of Korean medicine on LDH levels. In a study by Kim [33], 72 participants underwent integrative Korean medicine treatment using acupuncture, pharmacopuncture, Chuna manual therapy, herbal medicine and physical therapy, and the NRS, ODI, and EQ-5D-5L scores were significantly improved. In a study by Kim [34], Korean Medicine integrative treatment was performed in 46 participants, and statistically significant improvement was observed in the NRS scores for LBP, radiating leg pain, and Short-Form 36, another indicator of quality of life. Moreover, in a study by Jung [35], Korean Medicine integrative treatment was performed on 78 participants. Pre- and post-treatment L-spine MRI scans were compared, and it was confirmed that the herniated disc cross-section was reduced by approximately 47.8%.

This study used the SCB value to assess improvement in symptoms in the participants. In many studies, improvement is assessed by the minimal clinically important difference (MCID) [36,37]. The MCID value for patients with lumbar disc diseases is considered to be 1–2 points depending on the study [38], which is a “minimal” difference that is considered clinically significant. In addition, the MCID did not confirm sufficient therapeutic effects of MSAT. Accordingly, this study used SCB values to assess improvement in the symptoms of patients. In this study, the NRS for LBP was set to 2.5 points for the recovery criteria based on a previous study [39], which reported that the SCB of NRS for LBP in patients with chronic disc diseases was 2.5, and the results of an internal meeting of the study staff.

Few studies have evaluated the efficacy of MSAT for LDH [15,16,17,40]. A study by Shin [17] compared MSAT single treatment with NSAID injection treatment, and a study by Gang [16] did not have a control group, and thus there were differences between these and the current study. However, a study by Huh [40] that classified the participants into either the Korean Medicine integrative treatment group or the Korean Medicine integrative treatment group combined with MSAT is comparable to this study. In the above studies, pain during the early phase was much more severe in the MSAT group, but the proportion of pain reduction after treatment was greater. Thus, at the end of the treatment, the MSAT group experienced less pain. Overall, this study showed that MSAT can be utilized in patients with severe LDH and has great effects on pain reduction when combined with other integrative Korean medicine treatments. In addition, in this study, the patient’s condition improvement was maintained even after the end of treatment. However, since one study reported different results [15], further studies in this field should be performed. Meanwhile, the mechanisms of MSAT are still unclear; it has both the analgesic effect of acupuncture and the effect of changing the negative perception of pain [15]. Further studies on the mechanism are also needed.

There were some limitations to this study. As this was an observational study, the baseline characteristics between the two groups were not controlled for, which mandates the need for special attention when interpreting the results. Particularly, the assignment to the MSAT and non-MSAT groups was determined according to symptom severity and limitation of movement due to pain rather than random allocation. As a result, baseline values differed, especially for NRS for radiating pain, overall treatment duration, and number of treatments between the two groups. However, this may have resulted from evaluating treatment effects by reflecting the real world, and we tried our best to correct baseline differences using statistical techniques.

Another study limitation is the relatively low response rate. We tried to address this with several follow-up messages and small incentives; however, the final response rate was ~35%. Notably, the response rates of the MSAT and n-MSAT groups were almost the same, and the results of the final 152 subjects were similar to those of the 435 survey subjects. This suggests that the respondents were less biased. Lastly, the study contains interval-censored survival data; hence, using a general Kaplan–Meier curve may introduce bias.

This study aimed to investigate the add-on effect of MSAT in integrative Korean medicine treatments, not compare MSAT and integrative Korean medicine treatment. It is confirmed that add-on MSAT is also an effective and safe treatment for LDH in the study. Consequently, this study is significant in that it has been shown that integrative Korean medicine treatment combined with MSAT can be an effective treatment for LDH patients in the short and long term compared to integrative Korean medicine alone. We believe that well-designed RCTs are necessary to confirm our findings. The results of this study can provide helpful information for clinicians who treat patients with disc herniation in real clinical settings. Based on the results of this study, we expect that MSAT could be commonly used for treating patients with LDH and that patients who complain of severe pain can also receive better treatment.

## 5. Conclusions

This study demonstrated the efficacy and safety of integrative Korean medical treatment and MSAT in patients with LDH. Although there was a tendency for greater improvement in the MSAT group, there was no significant difference between the MSAT and n-MSAT group. Both groups showed overall significant improvement in pain and function. NRS for LBP and radiating leg pain and ODI decreased significantly after treatment. The results of this study can be used as helpful information for clinicians who treat patients with disc herniation in real clinical settings.

## Figures and Tables

**Figure 1 healthcare-10-02462-f001:**
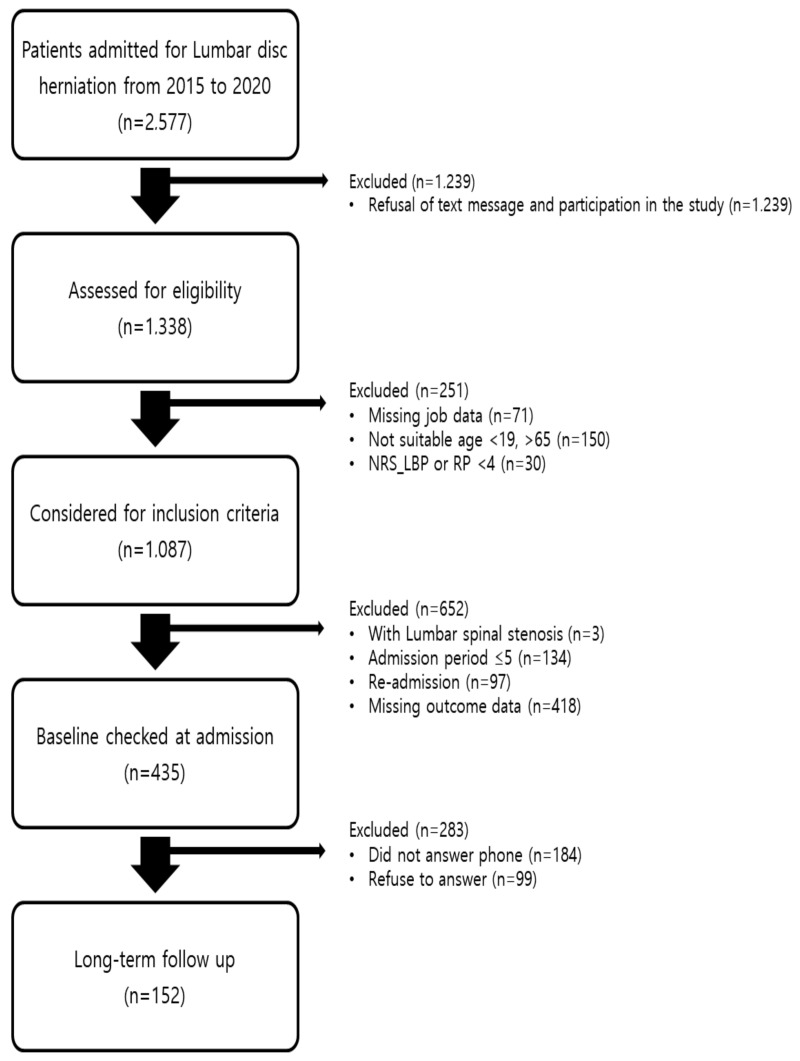
Flow chart. Abbreviations: Numerical Rating Scale (NRS); LBP, Low back pain (LBP); radiating leg pain (RP).

**Figure 2 healthcare-10-02462-f002:**
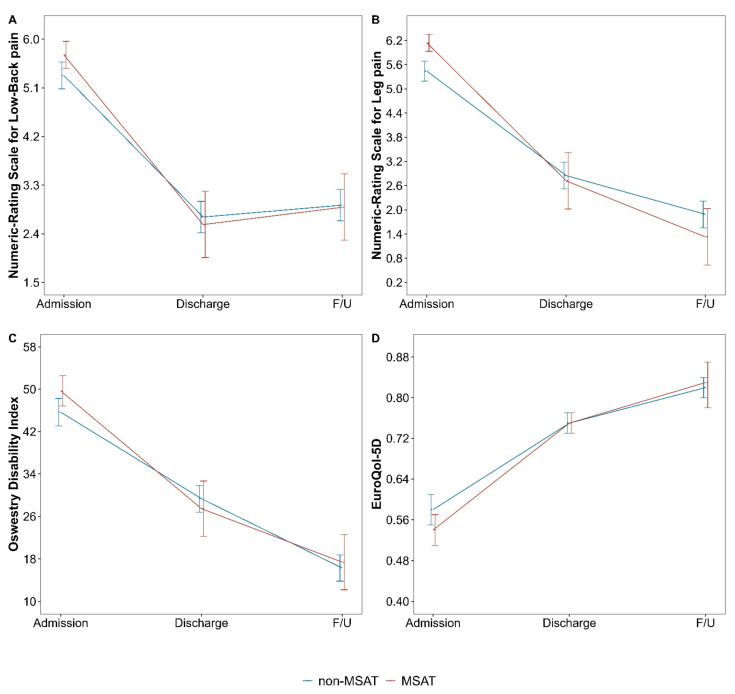
Changes in outcome at admission, discharge and follow-up in the MSAT and n-MSAT groups. Abbreviation: follow-up (F/U).

**Figure 3 healthcare-10-02462-f003:**
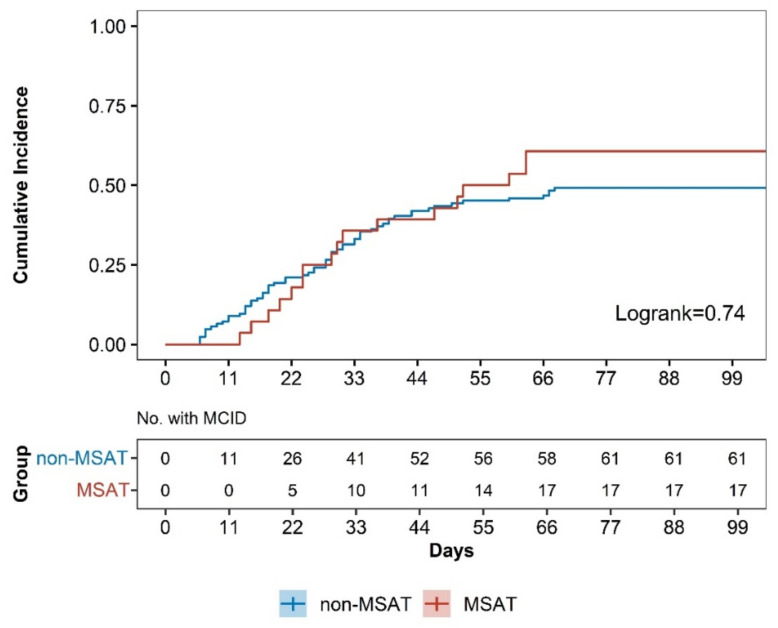
Survival analysis on improvement in NRS for LBP in the MSAT and n-MSAT groups. Abbreviation: minimal clinically important difference (MCID).

**Table 1 healthcare-10-02462-t001:** Basic characteristics of the study population.

Variable	Total (n = 152)	MSAT (n = 28)	Non-MSAT (n = 124)	*p*
Sex				0.434
Male	78 (51.32%)	12 (42.86%)	66 (53.23%)	
Female	74 (48.68%)	16 (57.14%)	58 (46.77%)	
Age	45.22 ± 12.67	48.82 ± 11.88	44.41 ± 12.74	0.096
Age group				0.616
19–29 years	20 (13.16%)	3 (10.71%)	17 (13.71%)	
30–39 years	34 (22.37%)	4 (14.29%)	30 (24.19%)	
40–49 years	35 (23.03%)	6 (21.43%)	29 (23.39%)	
50–59 years	38 (25.00%)	10 (35.71%)	28 (22.58%)	
≥60 years	25 (16.45%)	5 (17.86%)	20 (16.13%)	
Smoking				0.227
Smoking (past/now)	38 (25.00%)	4 (14.29%)	34 (27.42%)	
No smoking	114 (75.00%)	24 (85.71%)	90 (72.58%)	
Drinking				0.394
No	84 (55.26%)	18 (64.29%)	66 (53.23%)	
Yes	68 (44.74%)	10 (35.71%)	58 (46.77%)	
Job				0.98
No (including housewife/student)	64 (42.11%)	13 (46.43%)	51 (41.13%)	
Administration	14 (9.21%)	4 (14.29%)	10 (8.06%)	
Professional and related worker	17 (11.18%)	3 (10.71%)	14 (11.29%)	
Office worker	34 (22.37%)	5 (17.86%)	29 (23.39%)	
Service and salesperson	10 (6.58%)	2 (7.14%)	8 (6.45%)	
Agricultural, fishery, and skilled worker	1 (0.66%)	0 (0.00%)	1 (0.81%)	
Technicians and related skill workers	2 (1.32%)	0 (0.00%)	2 (1.61%)	
Device, machine operator and assembly worker	3 (1.97%)	0 (0.00%)	3 (2.42%)	
Simple worker	2 (1.32%)	0 (0.00%)	2 (1.61%)	
Solder	5 (3.29%)	1 (3.57%)	4 (3.23%)	
Visit type				1.000
No	2 (1.32%)	0 (0.00%)	2 (1.61%)	
Self-ambulation	146 (96.05%)	28 (100.00%)	118 (95.16%)	
With help and walker	3 (1.97%)	0 (0.00%)	3 (2.42%)	
S-CAR	1 (0.66%)	0 (0.00%)	1 (0.81%)	
Height	167.51 ± 8.94	165.86 ± 8.33	167.88 ± 9.06	0.281
Weight	70.08 ± 15.45	64.50 ± 12.79	71.34 ± 15.76	0.034
BMI*				
Mean	24.78 ± 3.98	23.26 ± 3.10	25.12 ± 4.09	0.025
				0.163
25<	64 (42.11%)	8 (28.57%)	56 (45.16%)	
≤25	88 (57.89%)	20 (71.43%)	68 (54.84%)	
Number of taking pain killers for LBP (n = 151)	3.29 ± 6.00	4.32 ± 6.31	3.06 ± 5.93	0.316
Number of injections related to steroids for LBP	0.78 ± 2.75	0.75 ± 1.46	0.78 ± 2.96	0.933
Back surgery and procedure				0.585
None	147 (96.71%)	28 (100.00%)	119 (95.97%)	
Yes	5 (3.29%)	0 (0.00%)	5 (4.03%)	
Length of hospital stay				
Mean	26.09 ± 17.56	34.93 ± 21.82	24.10 ± 15.89	0.018
Median	21.50 [14.00;34.00]	30.00 [19.50;49.00]	19.00 [13.00;32.00]	
Admission NRS LBP	5.40 ± 1.58	5.71 ± 1.58	5.33 ± 1.58	0.248
Admission NRS radiating leg pain	5.57 ± 1.56	6.14 ± 1.35	5.44 ± 1.58	0.032
Admission EQ-5D-5L	0.57 ± 0.19	0.54 ± 0.20	0.58 ± 0.19	0.283
Admission ODI	46.39 ± 16.72	49.69 ± 18.16	45.65 ± 16.36	0.249
Admission ROM flexion (n = 140)	70.21 ± 27.15	62.04 ± 32.79	72.17 ± 25.41	0.143
Admission ROM extension (n = 140)	16.04 ± 6.22	13.89 ± 7.64	16.55 ± 5.75	0.099
MRI findings				
Extrusion	67 (44.08%)	11 (39.29%)	56 (45.16%)	0.723
Protrusion	152 (100.00%)	28 (100.00%)	124 (100.00%)	
Sequestration	9 (5.92%)	0 (0.00%)	9 (7.26%)	0.212

Continuous variables were expressed as mean ± SD, and categorical variables were expressed as frequency and percentage. An independent *t*-test and chi-squared test or Fisher’s exact test were used to test continuous variables and categorical variables, respectively. Abbreviations: body mass index (BMI); European Quality of Life-5 Dimension-5 Level (EQ-5D-5L); magnetic resonance imaging (MRI); motion style acupuncture treatment (MSAT); Numerical Rating Scale (NRS); Oswestry Disability Index (ODI); range of movement (ROM); Stretcher Cart (S-CAR).

**Table 2 healthcare-10-02462-t002:** History of treatment during hospital stay.

Variable	Total (n = 152)	MSAT (n = 28)	Non-MSAT (n = 124)	*p*
H-MSAT				
Yes		16 (57.14%)		
Number of treatments performed		2.75 ± 2.74		
T-MSAT				
Yes		16 (57.14%)		
Number of treatments performed		8.50 ± 9.17		
Acupuncture and electroacupuncture				
Yes	152 (100.00%)	28 (100.00%)	124 (100.00%)	
Number of treatments	87.51 ± 60.91	116.14 ± 71.30	81.05 ± 56.65	0.006
Pharmacopuncture and bee venom acupuncture				0.212
Yes	143 (94.08%)	28 (100.00%)	115 (92.74%)	
Number of treatments performed	42.07 ± 31.37	63.46 ± 38.26	37.23 ± 27.54	0.002
Herbal medicine				0.115
Yes	80 (52.63%)	19 (67.86%)	61 (49.19%)	
Number of treatments performed	11.04 ± 17.86	15.57 ± 23.14	10.02 ± 16.38	0.237
Herbal medicine containing GCSB-5				0.765
Yes	132 (86.84%)	24 (85.71%)	108 (87.10%)	
Number of treatments performed	20.65 ± 17.63	27.18 ± 20.61	19.18 ± 16.63	0.03
Chuna manual therapy				0.74
Yes	135 (88.82%)	26 (92.86%)	109 (87.90%)	
Number of treatments performed	15.84 ± 11.70	19.96 ± 13.57	14.91 ± 11.09	0.039
Manipulative therapy				0.6
Yes	105 (69.08%)	21 (75.00%)	84 (67.74%)	
Number of treatments performed	9.09 ± 9.49	12.46 ± 10.69	8.32 ± 9.07	0.037
Physical therapy (including traction and extracorporeal shock wave therapy)				1.000
Yes	123 (80.92%)	23 (82.14%)	100 (80.65%)	
Number of treatments performed	55.18 ± 52.17	76.25 ± 69.27	50.42 ± 46.52	0.069

Status of treatment was represented as the number of participants who received treatments and percentage, and differences between two groups were analyzed using the chi-squared test or Fisher’s exact test. The number of treatments was expressed as the mean ± SD, and the difference between two groups was analyzed using the independent *t*-test.

**Table 3 healthcare-10-02462-t003:** Change in outcome at discharge and follow-up of participants.

	Admission (Baseline)	Discharge	Follow-Up
NRS for back pain	5.40 ± 1.58	2.68 ± 1.12	2.92 ± 2.09
Difference *		2.72 (2.45 to 2.98)	2.48 (2.22 to 2.74)
*p* value		<0.001	<0.001
NRS for leg pain	5.57 ± 1.56	2.83 ± 1.30	1.78 ± 2.36
Difference *		2.74 (2.44 to 3.04)	3.79 (3.49 to 4.09)
*p* value		<0.001	<0.001
ODI	46.39 ± 16.72	28.93 ± 13.71	16.47 ± 15.61
Difference *		17.41 (15.15 to 19.67)	29.90 (27.68 to 32.12)
*p* value		<0.001	<0.001
EQ5D	0.57 ± 0.19	0.75 ± 0.10	0.82 ± 0.14
Difference *		−0.18 (−0.20 to −0.16)	−0.25 (−0.27 to −0.23)
*p* value		<0.001	<0.001

Outcome values at each time point were expressed as the mean ± SD. * Change from baseline in each outcome was estimated using linear mixed model with adjusted baseline values, and was expressed as the estimated mean and 95% CI.

**Table 4 healthcare-10-02462-t004:** Difference in outcome at each time point between two groups.

		Admission (Baseline)	Discharge	Follow-Up
NRS LBP	MSAT	5.71 ± 1.58	2.57 (1.96 to 3.19)	2.89 (2.28 to 3.51)
	Non-MSAT	5.33 ± 1.58	2.71 (2.42 to 3.00)	2.93 (2.64 to 3.22)
	Difference *		0.14 (−0.54 to 0.82)	0.03 (−0.65 to 0.71)
	*p* value		0.693	0.924
NRS RP	MSAT	6.14 ± 1.35	2.72 (2.02 to 3.42)	1.33 (0.63 to 2.03)
	Non-MSAT	5.44 ± 1.58	2.85 (2.52 to 3.18)	1.89 (1.55 to 2.22)
	Difference *		0.13 (−0.64 to 0.91)	0.56 (−0.22 to 1.33)
	*p* value		0.738	0.159
ODI	MSAT	49.69 ± 18.16	27.46 (22.26 to 32.66)	17.40 (12.19 to 22.60)
	Non-MSAT	45.65 ± 16.36	29.34 (26.82 to 31.86)	16.29 (13.82 to 18.76)
	Difference *		1.89 (−3.90 to 7.67)	−1.11 (−6.87 to 4.66)
	*p* value		0.522	0.706
EQ5D	MSAT	0.54 ± 0.20	0.75 (0.71 to 0.80)	0.83 (0.78 to 0.87)
	Non-MSAT	0.58 ± 0.19	0.75 (0.73 to 0.77)	0.82 (0.80 to 0.84)
	Difference *		0.00 (−0.06 to 0.05)	−0.01 (−0.06 to 0.04)
	*p* value		0.866	0.737

Each outcome baseline value by group was expressed as the mean ± SD. Outcomes at discharge and follow-up were expressed as the estimated values by the linear mixed model with adjusted baseline and were expressed as estimated mean and 95% CI. * Difference is a change in indicators in both groups from baseline and was estimated using linear mixed model.

**Table 5 healthcare-10-02462-t005:** Logistics for NRS of LBP.

	Univariate	Multivariate
	OR (95% CI)	OR (95% CI)
Intercept		
Baseline outcome	3.01 (2.07–4.40)	3.42 (2.24–5.21)
MSAT (ref = non-MSAT)		
MSAT	1.20 (0.50–2.88)	0.59 (0.18–2.01)
Sex (ref = Male)		
Female	0.65 (0.33–1.27)	0.27 (0.08–0.91)
Age (ref = age < 50 years)		
Age ≥ 50 years	1.70 (0.85–3.39)	2.06 (0.74–5.74)
BMI (ref = BMI < 25)		
BMI > 25	1.09 (0.56–2.14)	1.24 (0.45–3.42)
Smoking (ref = no)		
Smoking (past/current)	0.93 (0.43–1.99)	0.82 (0.25–2.69)
Drinking (ref = no)		
Yes	1.14 (0.59–2.24)	0.97 (0.36–2.62)
Job (ref = none)		
Employed	1.03 (0.53–2.02)	0.68 (0.23–2.03)
Length of hospital stay (ref = <15 days)		
Length of hospital stay: 15–21 days	1.07 (0.43–2.69)	1.65 (0.47–5.81)
Length of hospital stay: 22–28 days	1.58 (0.53–4.70)	2.96 (0.63–13.84)
Length of hospital stay: >28 days	2.32 (0.99–5.45)	3.42 (0.90–12.96)
MRI finding		
MRI finding (presence of extrusion): Yes	1.39 (0.71–2.73)	1.05 (0.41–2.68)
AUC		0.88

Abbreviations: area under the curve (AUC); odds ratio (OR).

## Data Availability

The data presented in this study are available on request from the corresponding author. The data are not publicly available due to privacy/ethical restrictions.

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
