# Peer review of "Long-Term Follow-Up of Intensive Integrative Treatment including Motion Style Acupuncture Treatment (MSAT) in Hospitalized Patients with Lumbar Disc Herniation: An Observational Study"

_healthcare, 2022, doi:10.3390/healthcare10122462_

Round 1

Reviewer 1 Report

Dear Authors,

Overall I think this is an interesting study and the idea of MSAT as a treatment option for pain seems worthy to investigate. Below are my comments;

Lines 547-549 below, I do not feel need to be added in the limitations section. This section should speak mainly to those things that could have been improved in the study, or areas that fall short. I don't think adding content to justify use of MSAT or other Medicine is needed. 

Korean medicine is commonly used, especially in the musculoskeletal system, and many relevant studies have been conducted [41, 42]. However, there are few studies on MSAT that have combined acupuncture with daoyin therapy [15, 40, 43].

Also, lines 549 to 554 I feel are unnecessary in the limitations section. Providing further information in the limitations section to justify results seems a little bias. In addition, there is no discussion that I saw in the text discussing bias from the authors perspective.

This study conducted a large-scale survey at three hospitals located in various regions of Korea and confirmed the efficacy and safety of integrative Korean medical treatment and MSAT in patients with LDH. In particular, the study demonstrated that integrative Korean medicine treatment combined with MSAT has great effects on reducing pain compared to integrative Korean medicine treatment alone.

I feel the conclusion needs to be more robust, at least a few more sentences to provide a better summary of the reader.

While this review did follow STROBE guidelines, I feel the discussion section would be improved by adding a small section discussing the proposed mechanisms of MSAT. Speaking to clinical benefit is important, however, a discussion on proposed mechanisms with supporting references would make for a stronger discussion and be more interesting to readers.

Reviewer 2 Report

1. The Abstract had no conclusion. In addition, a more detailed presentation of results and statistics was needed. If it was because of the word limit, the description of the outcome measures should be reduced, and the results and conclusion should be supplemented more.

2. The NRS for ‘low back pain’ and the NRS for ‘leg pain’ should be distinctly described throughout the abstract and manuscript.

3. Only a few of 152 LDH patients were treated with MSAT, but the clinical criteria for special MSAT treatment need to be explained. It was insufficient to explain simply by assuming that it would have been performed on patients with high severity. It is necessary to describe in detail how MSAT is performed clinically based on the criteria for determining severity. Since the MSAT and non-MSAT groups were not randomly divided, these criteria may bias the results.

4. It is necessary to present data on whether the 152 participants finally selected could represent the 2,577 patients. In addition, if the outcome values before and after hospitalization were collected through EMR, it is considered an appropriate research method to collect data regardless of the patient's refusal to participate or non-response. In the case of data collected through the existing medical record, informed consent can be waived according to the following criteria.

“If consent is waived because it is not practicable to obtain consent from large numbers of patients for a retrospective chart review study, generally it also will not be appropriate to attempt to contact those patients to tell them about the study retrospectively.”

5. Table 1 showed statistical differences in critical indicators, which required interpretation.

6. In Table 2, there were some statistical differences in the number of treatments, but an explanation was needed considering the difference in hospitalization period in Table 1. Additionally, information on the number of the MSAT treatment should be presented in a table or manuscript.

7. To perform survival analysis, it is necessary to accumulate data through continuous follow-up for a certain period. In this study, data were collected only three times: before hospitalization, at discharge, and at the time of F/U, but how survival analysis is possible was questionable.

8. “Result” described “For all indicators, the differences in the changes between the groups were not significant (Table 4, Figure 2).”, but “Discussion” described “In addition, the difference in the change in outcomes after treatment was more significant in the MSAT group. / In particular, the study demonstrated that integrative Korean medicine treatment combined with MSAT has great effects on reducing pain compared to integrative Korean medicine treatment alone”, which contradicted the content. In addition, it was necessary to explain what “*” of Difference* in Table 4 meant.

9. “There were no adverse events (AEs) identified when reviewing the medical records.” In this sentence, it was necessary to confirm that there was no adverse events to all integrative Korean medical treatments. The collection method for AEs was not in “2. Methods”.

10. Overall, the contents of “integrative Korean medical treatment” and “MSAT” were mixed. It would be appropriate to clearly separate the sections or report them as separate papers.

Round 2

Reviewer 2 Report

The comments suggested in the previous review have been well improved.